# Economic evaluation of dialysis treatment in end-stage renal disease patients with fluid and sodium overload: Evidence from a randomized controlled trial in Thailand

**Sitaporn Youngkong**[1,2], **Panida Yoopetch**[1,3], **Montarat Thavorncharoensap**[1,2], **Montira Assanatham**[1,4], **Usa Chaikledkaew**[1,2]*, **Suchai Sritippayawan**[5]

**1** Mahidol University Health Technology Assessment (MUHTA) Graduate Program, Mahidol University, Bangkok, Thailand, **2** Social and Administrative Pharmacy Division, Department of Pharmacy, Faculty of Pharmacy, Mahidol University, Bangkok, Thailand, **3** Department of Pharmacy, Rajavithi Hospital, Bangkok, Thailand, **4** Division of Nephrology, Department of Medicine, Faculty of Medicine Ramathibodi Hospital, Mahidol University, Bangkok, Thailand, **5** Division of Nephrology, Department of Medicine, Faculty of Medicine Siriraj Hospital, Mahidol University, Bangkok, Thailand

* usa.chi@mahidol.ac.th

**Data availability statement:** Data cannot be shared publicly because our study involved end-stage renal disease patients with fluid and

## Abstract

Given the lack of cost-effectiveness information, continuous ambulatory peritoneal dialysis (CAPD) with icodextrin (CAPD+ICO) has not yet been included in the Universal Health Coverage (UHC) scheme. This study aimed to evaluate the cost-utility of dialysis treatments for end-stage renal disease (ESRD) patients with fluid and sodium overload, comparing CAPD+ICO and automated peritoneal dialysis (APD) against glucose-based CAPD. A Markov model was applied to evaluate lifetime costs and health outcomes from a societal perspective. Data, including transitional probabilities, direct medical and non-medical costs, and utilities, were collected from randomized controlled trials conducted across 16 hospitals in various regions of Thailand. Compared to glucose-based CAPD, the incremental cost-effectiveness ratio (ICER) for CAPD+ICO was 908,440 THB (26,082 USD) per quality-adjusted life year (QALY) gained, while APD was dominated, incurring higher costs and yielding fewer QALYs. The results indicated that glucose-based CAPD had a 90% probability of being the most cost-effective option from a societal perspective, based on Thailand's willingness-to-pay (WTP) threshold of 160,000 THB (4,603 USD) per QALY gained. Therefore, CAPD+ICO is not considered a good value for money, requiring an additional annual budget of approximately 58 million THB (2 million USD). These findings provide important economic evaluation evidence to support policy decision-making alongside clinical effectiveness and equity considerations in guiding future UHC benefit package decisions for dialysis modalities among ESRD patients with fluid and sodium overload in Thailand.

sodium overload, and the dataset contains sensitive personal and clinical information. Due to ethical restrictions and participant confidentiality concerns, we are unable to publicly share the full dataset, even in anonymized form, as per the conditions of participant consent and institutional review board approval. However, data are available upon reasonable request to qualified researchers, subject to institutional review and approval. Requests may be directed to the Human Research Protection Unit, Faculty of Medicine Siriraj Hospital, Mahidol University, at Room 210, 2nd Floor, His Majesty the King's 80th Birthday Anniversary 5th December 2007 Building, 2 Wang Lang Road, Bangkoknoi, Bangkok 10700, or via email at siethics@mahidol.ac.th.

**Funding:** This research project has been funded by the Health Systems Research Institute (HSRI). The findings, interpretations and conclusions expressed in this article do not necessarily reflect the views of the aforementioned funding agency. The funders had no role in study design, data collection and analysis, decision to publish, or preparation of the manuscript.

**Competing interests:** The authors have declared that no competing interests exist.

## Introduction

In the context of renal replacement therapy (RRT), the focus has shifted from merely extending life to also enhancing the quality of life [1–3]. Several factors affect the quality of life for end-stage renal disease (ESRD) patients undergoing RRT. These include physical symptoms such as fatigue, itching, body pain, cramps, dizziness, and breathing difficulties [4], as well as hemoglobin levels, which can significantly impact patient well-being [5]. In Thailand, the number of ESRD patients requiring RRT continues to rise, with three main treatment options, i.e., continuous ambulatory peritoneal dialysis (CAPD), hemodialysis (HD), and kidney transplantation (KT).

Thailand's National Health Security Office (NHSO) has expanded dialysis services under Peritoneal Dialysis (PD) First Policy, officially implementing CAPD as a primary treatment since January 1, 2008 [6]. However, some patients experience inadequate PD, particularly those with larger body sizes or volume overload. This occurs when the peritoneal membrane becomes damaged from prolonged exposure to glucose-based dialysis solutions or infections [7]. Such patients are at risk for fluid overload, which can lead to pulmonary edema, heart complications, and eventually death if untreated [8]. Although HD can alleviate these issues, some patients are unable to transition to HD due to physical limitations or logistical challenges, increasing the risk of fatal heart disease caused by fluid overload [8].

Patients with inadequate PD may benefit from using hypertonic solutions, which can help manage fluid overload [9]. However, this approach carries risks, particularly for diabetic patients, as it can lead to uncontrolled high blood sugar levels. Additionally, long-term exposure to elevated glucose concentrations can cause deterioration of the peritoneal membrane, increasing the risk of developing encapsulating peritoneal sclerosis (EPS), a potentially fatal condition [9]. Therefore, while hypertonic solutions can offer short-term benefits, careful monitoring and management are essential to mitigate these risks [9]. In countries with fewer financial constraints, patients experiencing inadequate PD are often transitioned to glucose polymer-based solutions like icodextrin or to automated peritoneal dialysis (APD). These alternatives can more effectively remove excess fluid and sodium, thereby improving the management of fluid overload.

Icodextrin, in particular, offers the advantage of reducing the risk of hyperglycemia associated with traditional glucose-based solutions, while APD provides a more efficient dialysis process, allowing for greater flexibility and potentially better patient outcomes [10]. APD enhances dialysis efficiency by utilizing a machine to facilitate more frequent exchanges, allowing for increased dialysate volumes. In contrast, CAPD is manually limited to approximately five exchanges per day. This increased efficiency makes APD particularly advantageous for pediatric patients and working adults, as it can be performed overnight while they sleep, freeing up time during the day for other activities. Additionally, some studies indicate that patients on APD report a better quality of life in specific areas, such as having more time for personal and professional pursuits compared to those undergoing CAPD [11].

However, the costs associated with icodextrin solutions and APD in Thailand are substantially higher than those for CAPD. Specifically, icodextrin is priced at

approximately 700 Thai baht (THB) (20 United States dollar, USD) per bag per day, about twice the cost of CAPD, while APD can be four times more expensive, costing around 50,000 THB (1,439 USD) per month. To date, no cost-effective studies have been conducted comparing CAPD using icodextrin or APD with glucose-based CAPD for ESRD patients with fluid and sodium overload. There is currently a lack of cost-effectiveness and budget impact information for icodextrin and APD among Thai patients with ESRD, particularly concerning their effects on fluid and sodium overload. This absence of data creates significant barriers to access for these potentially beneficial treatments. Therefore, this study aimed to evaluate the cost-utility and budget impact of CAPD+ICO and APD compared to glucose-based CAPD. The results of this study will provide critical insights to inform decision-making regarding cost-effectiveness of dialysis treatments within Thailand's healthcare system.

## Methods

### Study design

A cost-utility analysis using a Markov model with one-year cycle length was conducted to evaluate lifetime costs and health outcomes of CAPD+ICO and APD compared with glucose-based CAPD based on a societal perspective in Thailand. The target population consisted of patients initiating treatment at the age of 55 years, specifically focusing on those experiencing fluid and sodium overload and undergoing PD. Data, including transitional probabilities, direct medical and non-medical costs, and utilities, were specifically collected from a randomized controlled trial (RCT) conducted across 16 hospitals in various regions of Thailand [12]. S1 Table presents baseline characteristics of study participants.

### Ethics approval

This study was approved by the Siriraj Institutional Review Board, Faculty of Medicine Siriraj Hospital, Mahidol University (COA no. Si 764/2020). Since we obtained data from published studies, the ethics committee waived the requirement for informed consent. All methods were carried out in accordance with International Guidelines for Human Research Protection such as the Declaration of Helsinki, the Belmont Report, CIOMS Guidelines and the International Conference on Harmonization in Good Clinical Practice (ICH-GCP).

### Economic model

Fig 1 demonstrates the Markov model used in this study. To ensure the model closely resembled real-world practices, it allowed patients to switch between the PD methods. Patients could transition to other RRT such as HD or KT and switch

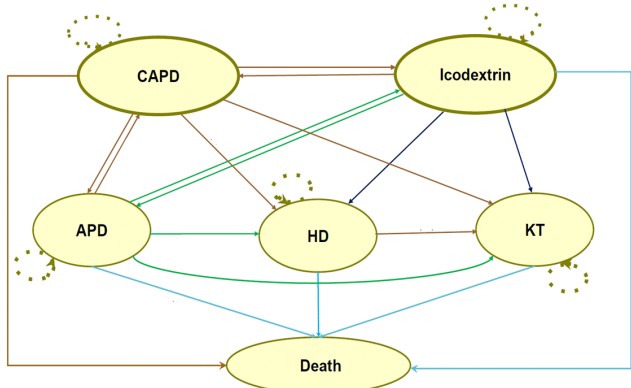

**Fig 1. Markov model.**

between RRT throughout their lifetime. After transitioning from PD to HD, patients could further move to KT, death, or remain in their current health state. For those who received a KT, they could either stay in the KT state or transition to death. However, temporary transitions, such as a CAPD patient temporarily switching to HD due to a peritoneal infection and then returning to CAPD post-recovery, were not considered in the model. Microsoft Excel was used to develop the Markov model and perform the data analysis.

## Model parameters

Data on transitional probabilities, clinical outcomes, costs and utility values for the first, second and third years were obtained from a multi-center, open labeled RCT [12]. The RCT study employed a two-stage randomization process to minimize bias and ensure comparability across treatment groups. Block randomization was used within each participating hospital to balance treatment allocation, while stratified randomization accounted for key prognostic factors, specifically duration of PD treatment (≤3 years vs. >3 years) and the type of APD machine used (Baxter vs. Fresenius Medical Care).

Eligible participants were adults aged 18 years or older enrolled in Thailand's UHC scheme and had been receiving CAPD for more than three months at Ministry of Public Health or public university hospitals across various regions of Thailand. Inclusion criteria required patients to use 2-liter per cycle PD solutions for three to five cycles daily, including at least two cycles with 2.5% dextrose or at least one cycle with 4.25% dextrose. Additionally, participants had to exhibit at least one clinical indicator of fluid overload, i.e., edema, poorly controlled hypertension (systolic blood pressure (SBP) >140 mmHg or diastolic blood pressure (DBP) >90 mmHg), or a history of congestive heart failure (CHF) symptoms within the past year. Patients also had to be able to communicate in Thai to complete health-related quality of life (HRQoL) assessments.

Exclusion criteria included edema from non-fluid-related causes such as catheter malfunction (e.g., malposition or omental wrapping) or suspected dialysate leakage confirmed by a rapid fill-and-drain test (discrepancy ≥ 200 mL between infused and drained volumes). Patients were also excluded for edema due to medication side effects, hypothyroidism, liver cirrhosis, or malnutrition (defined as serum albumin <2 g/dL within the past two months). Additional exclusions included peritonitis within the previous month, baseline SBP < 100 mmHg, current diagnosis of sepsis, acute coronary syndrome, CHF, or stroke, prior use of icodextrin or home-based APD, pregnancy, terminal cancer or life expectancy less than six months, or any change in healthcare coverage during the study period.

Patients were randomized to one of three PD modalities, i.e., CAPD, CAPD+ICO, or APD. All patients received monthly treatment adjustments aimed at resolving fluid and sodium overload defined as resolution of edema, stable BP (100–140/<90 mmHg) without dizziness, and absence of CHF symptoms, and achieving adequate solute clearance (total weekly Kt/V urea >1.7 or normalized creatinine clearance >45 L/1.73 m$^2$) within two months. CAPD patients had their number of exchanges and glucose concentrations adjusted as needed. In the CAPD+ICO group, one low-glucose exchange was replaced with icodextrin, with similar adjustments to remaining exchanges. APD patients started with a nighttime intermittent PD regimen using 10 liters of 2.5% dextrose with adjustments up to 20 liters as necessary, and day dwell exchanges were added if fluid overload or inadequate clearance persisted. Tidal PD was permitted in cases of inflow/out-flow discomfort or suboptimal drainage.

A total of 180 patients were enrolled between November 16, 2020 – March 8, 2021, and were equally allocated into CAPD, CAPD+ICO, and APD groups (60 patients each) [12]. Participants were recruited from 16 hospitals across all regions of Thailand, with most hospitals. enrolling 12 patients (4 per group), except for two that enrolled 6 patients each. Patients were followed up every two months to assess their clinical outcomes and determine transitional probabilities, including death, changes in dialysis modality, or transition to KT, over a three-year period [12]. Direct medical cost data were retrieved from patients' medical records, while direct non-medical costs and utility data were collected through patient interviews using a cost data collection form and EQ-5D-5L questionnaires, respectively [12].

**Transitional probabilities.** S2 Table presents all parameters used in this study. Data including clinical outcomes and transitional probabilities between health states were derived from RCT conducted over a three-year period. Clinical outcomes included the rates of peritonitis, mortality, adverse events, and transitions between dialysis types among the three patient groups, i.e., CAPD, APD, and CAPD+ICO. Transitional probabilities for health state transitions in the first, second, third year were obtained from the RCT study [12], while probabilities beyond the third year were assumed to remain constant for the patient's lifetime.

**Costs.** Direct medical costs encompass all resources used in health interventions, including those for diagnosis, treatment, follow-up rehabilitation, and end-of-life care. Based on the previous RCT study [12], average expense data from patients undergoing RRT at the first, second and third year after enrollment were analyzed to reflect the actual costs incurred by those treated with CAPD, APD, and CAPD+ICO. We did not have access to information that could identify individual participants during or after data collection. These costs include both out-of-pocket payments made by patients and those covered by the UHC scheme, along with expenses related to adverse effects. In severe cases, costs also accounted for hospital stays and expenditures on life-saving measures. To accurately represent the true costs of treatment, incurred expenses were converted from charges (amounts recorded in the hospital system) using a cost-to-charge ratio of 1.63.

Moreover, direct non-medical costs refer to out-of-pocket expenses incurred by patients or their families for goods and services that extend beyond medical care charges. These can include expenses for food, amenities, and wages for caregivers. In situations where formal caregivers are not hired and family members assist the patient, these contributions are classified as informal care costs. In this study, the costs associated with informal care were evaluated using the human-capital method. This method calculates the average gross national income (GNI) per capita annually. The assessment involves multiplying the total number of caregiving hours by an adjusted hourly wage based on GNI, then dividing this figure by 52 weeks in a year and a standard working time of 48 hours per week. From the previous RCT study [12], we obtained direct non-medical cost data of patients who were interviewed using a cost data collection form to ascertain the average caregiving time for each patient over a specified period, subsequently averaging this information for each treatment option. All costs were adjusted to 2023 values to ensure consistency within the same fiscal year, using the Consumer Price Index (CPI) for healthcare services and medications. Costs were then converted from THB to USD based on the 2023 exchange rate (1 THB = 0.02877 USD) [13].

**Utility.** According to the previous RCT study, utility data were collected using the Thai version of the EQ-5D-5L questionnaire through interviews with patients undergoing RRT [12]. This data analysis aimed to reflect the HRQoL of patients treated with the three PD methods: CAPD, APD, and CAPD+ICO. By incorporating utility values derived from patient responses, the study intended to provide a comprehensive assessment of quality-adjusted life years (QALYs) associated with each treatment option.

## Result presentation

Future costs and outcomes, measured in QALYs, were discounted at a rate of 3%, in accordance with the Thai health technology assessment (HTA) guidelines [14], while budget impact analysis did not apply any discounting. The results were presented as an incremental cost-effectiveness ratio (ICER), calculated by dividing the difference in costs by a difference in QALYs. If the ICER is less than the Thai societal willingness to pay (WTP) threshold, set at 160,000 baht (4,603 USD) per QALY gained, it is considered a cost-effective option [15].

## Uncertainty analysis

To account for the uncertainty of various input variables, one-way sensitivity analysis, probabilistic sensitivity analysis (PSA), and scenario analysis were conducted. One-way sensitivity analysis involves varying one variable at a time while keeping all other variables constant, in order to assess their impact on changes in ICER.The results were presented using

a Tornado diagram, which highlights the variables with the greatest influence on the ICER. In addition, PSA allows for the simultaneous variation of multiple important variables according to predefined distribution characteristics. It utilizes Monte Carlo simulation techniques to repeatedly simulate the results of the cost-effectiveness analysis, running the model 1,000 times. The results were presented using cost-effectiveness planes and cost-effectiveness acceptability curves.

As the RCT data only covered the first three years, we assumed a constant mortality rate beyond year 3 based on the average annual mortality observed in years 1–3. A scenario analysis was conducted to assess the impact of increasing the probability of death from year 3 onward (ranging from 0% to 100%) on the ICER values of CAPD+ICO and APD compared to glucose-based CAPD. Additionally, a scenario analysis was conducted using age-specific mortality rates and EQ-5D–based health utility values for patients receiving CAPD and APD, derived from an external study conducted in Singapore [16], to assess the robustness and external validity of the model. These inputs were applied to the same Markov model structure used in the base case. The purpose of this scenario analysis was to examine whether the model's conclusions would remain consistent when applying alternative input data from a comparable regional setting.

### Model validation

To assess the validity of the model outputs, both face validation and external validation techniques were employed. Face validation was performed through consultation with clinical experts in nephrology and health economists to verify the model structure, assumptions, key inputs, and preliminary results ensuring clinical plausibility and alignment with real-world treatment pathways in Thailand. For external validation, model-predicted age-specific mortality rates across dialysis modalities (CAPD, APD, and CAPD+ICO) were compared with observed mortality data from a recently published cohort study conducted in Thailand by Thuanman et al. (2024), which investigated the survival of 1,858 ESRD patients on PD who were hospitalized due to fluid overload between January 2008 and December 2018 [17]. This comparison aimed to assess whether the modeled mortality trends were consistent with empirical real-world evidence.

### Budget impact analysis

The budget impact analysis evaluates the financial implications of adopting different treatment methods, i.e., CAPD, APD, and CAPD+ICO for ESRD for patients experiencing fluid and sodium overload based on the governmental perspective covering only direct medical costs. To estimate the total number of ESRD patients with fluid and sodium overload, the analysis used data from the UHC scheme in Thailand. The budget impact was calculated by multiplying the estimated number of ESRD patients by the annual treatment costs per patient for each dialysis method and projected over a 10-year period.

## Results

### Cost-utility analysis

Table 1 displays the results of cost-utility analysis. The study revealed significant findings regarding the cost-effectiveness of different dialysis treatments for ESRD patients. From a societal perspective, the lifetime total costs were 3,429,977 THB (98,680 USD) for CAPD+ICO, 2,623,402 THB (75,475 USD) for APD, and 2,487,161 THB (71,556 USD) for CAPD. In terms of QALYs, CAPD+ICO, APD, and CAPD yielded 3.80 QALYs, 2.54 QALYs, and 2.76 QALYs, respectively. Compared to CAPD, the ICER of CAPD+ICO was 908,440 THB (26,082 USD) per QALY gained from the societal perspective. In contrast, the ICER for APD was negative, indicating that it increased total costs while decreasing QALYs, thus rendering it a dominated option.

### Uncertainty analysis

**One-way sensitivity analysis.** The results of one-way sensitivity analysis are illustrated in the Tornado diagram (Fig 2). The X-axis shows the percentage change in ICER of the CAPD+ICO method compared to CAPD. The diagram

**Table 1. Cost-utility analysis results.**

| Results in THB (USD) | CAPD | CAPD+ICO | APD |
|---|---|---|---|
| Total cost | 2,487,161 (71,556) | 3,429,977 (98,680) | 2,623,402 (75,475) |
| Total LYs | 3.56 | 4.92 | 3.42 |
| Total QALYs | 2.76 | 3.80 | 2.54 |
| Incremental costs | | 942,816 (27,125) | 136,241 (3,920) |
| Incremental LYs | | 1.36 | −0.14 |
| Incremental QALYs | | 1.04 | −0.22 |
| ICER per LY gained | | 692,532 (19,945) | Dominated[a] (−956,969) (−27,998) |
| ICER per QALY gained | | 908,440 (26,082) | Dominated[a] (−619,738) (−17,817) |

[a]Dominated in this context refers to higher costs but lower LYs or QALYs. LYs, life years; QALY, quality adjusted life years; ICER, incremental cost-effectiveness ratio; USD, United States dollar; THB, Thai baht.

highlighted the top five variables that most influence the ICER: 1) the cost of outpatient visits per week for the HD group, 2) the number of weeks per year receiving dialysis treatment across all modalities, 3) the number of inpatient visits per year for the icodextrin group, 4) the number of inpatient visits per year for the CAPD group, and 5) the cost of icodextrin. In contrast, the ICER for the APD method compared to CAPD (Fig 3) also reflects changes from the base case, indicating the top five influential variables on the ICER: 1) the probability of death in the CAPD group in year 2, 2) the probability of

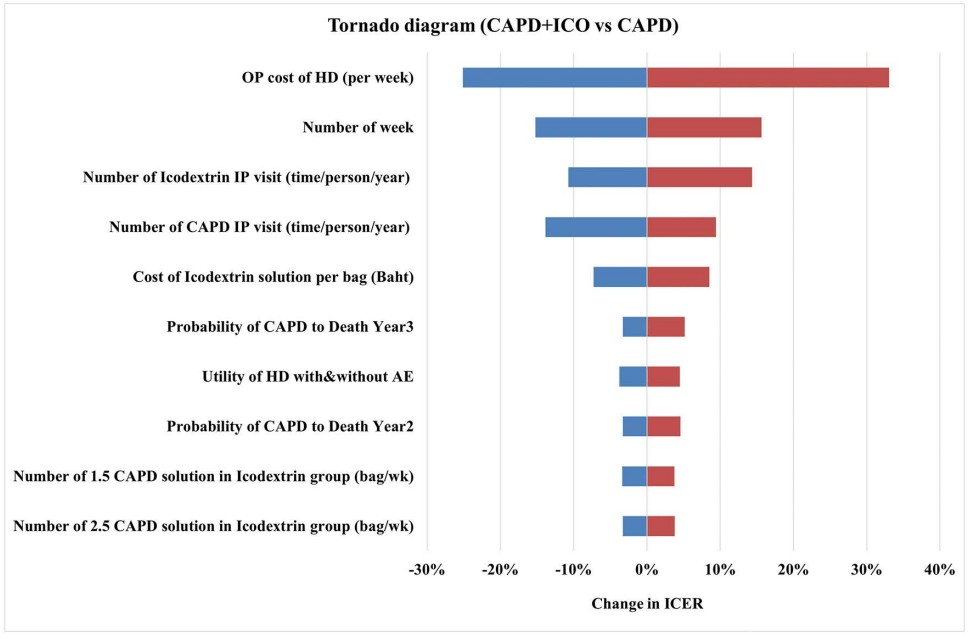

**Fig 2. Tornado diagram of CAPD + ICO method.**

death in the APD group in year 2, 3) the probability of death in the CAPD group in year 3, 4) the number of inpatient visits per year for the APD group, and 5) the probability of death in the APD group in year 3.

**Scenario analysis.** To assess the impact of increasing the probability of death from year 3 onward (ranging from 0% to 100%), we found that for CAPD+ICO, both LYs and QALYs gained increased slightly across all scenarios, while the incremental cost remained relatively stable (S3 Table). The ICER values ranged from 908,440 baht (26,136 USD) per QALY gained at 0% increase in mortality to 881,825 baht (25,370 USD) per QALY gained at a 100% increase, indicating a modest improvement in cost-effectiveness as mortality assumptions became more conservative. The intervention remained consistently not cost-effective across all mortality scenarios at the Thai societal WTP of 160,000 baht (4,603 USD) per QALY gained.

For APD, the intervention was associated with negative incremental QALYs up to a 60% increase in mortality, reflecting inferior effectiveness compared to CAPD using glucose solution under those assumptions. At 60% and beyond, incremental QALYs turned positive, but the ICER values were substantially higher, ranging from 2.39 to 9.63 million THB per QALY gained. This indicates that APD remained not cost-effective under most scenarios at the current WTP threshold. Overall, the scenario analysis demonstrated that the relative cost-effectiveness ranking of the three dialysis modalities remained stable across a wide range of long-term mortality assumptions, supporting the robustness of the model outcomes.

In addition, scenario analysis was also conducted using age-specific mortality rates and EQ-5D–based health utility values from Yang et al.'s study in Singapore to assess the robustness of the model's conclusions. Compared to the base case, the ICER for CAPD+ICO versus glucose-based CAPD increased substantially from 908,440 to 6,734,400 THB per QALY gained, reflecting a significant reduction in incremental QALYs (from 1.04 to 0.14), while incremental costs remained the same (S4 Table). This suggests that CAPD+ICO is considerably less cost-effective under these external assumptions. APD remained dominated in both the base case and the scenario analysis, with higher costs and

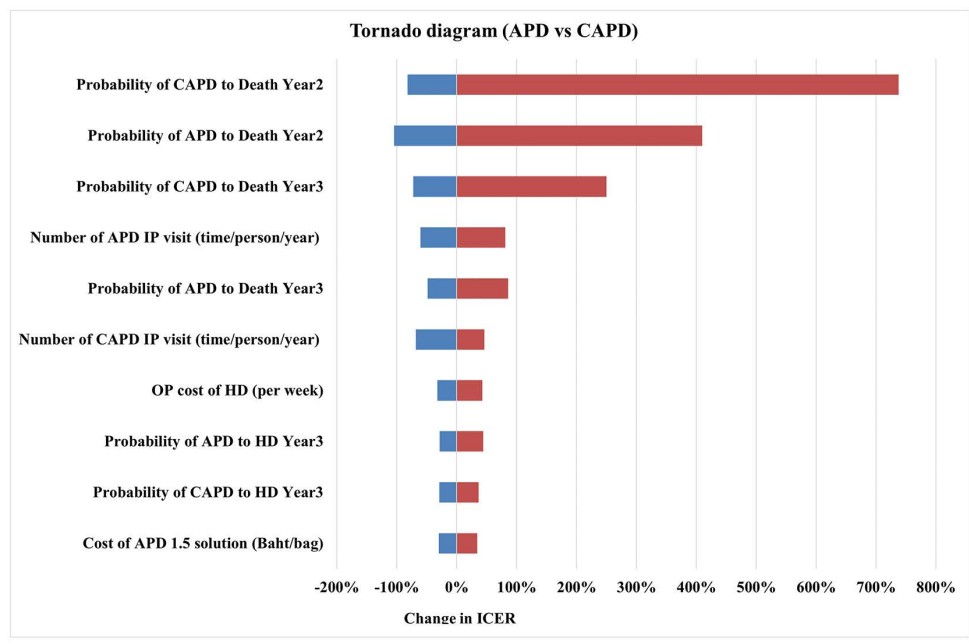

**Fig 3. Tornado diagram of APD method.**

lower QALYs compared to glucose-based CAPD. Notably, the loss in QALYs with APD was more noticeable under the Singapore-based inputs (−0.48 vs. −0.22), further diminishing its cost-effectiveness.

**Probabilistic sensitivity analysis.** The results of the PSA from a societal perspective are presented in the cost-effectiveness acceptability curve (Fig 4). The X-axis represents the cost-effectiveness threshold or WTP (in THB per QALY gained), while the Y-axis indicates the probability that each option is cost-effective at different WTP thresholds. It was found that at a WTP of 160,000 THB (4,603 USD) per QALY gained, neither APD nor CAPD+ICO was economically cost-effective. Glucose-based CAPD had the highest probability of being cost-effective (87%), followed by APD (13%). When applying mortality and utility parameters from Yang et al.'s study, CAPD remained dominant with a 97% probability of cost-effectiveness, reinforcing the robustness of our conclusions across parameter variations (Fig 5).

Furthermore, the results of the PSA are displayed on the cost-effectiveness plane (Fig 6). The Y-axis represents the incremental cost, i.e., the difference in total lifetime costs, while the X-axis represents the incremental QALYs, i.e., the difference in total QALYs between CAPD+ICO or APD and CAPD. The summary of the findings indicates that from 1,000 iterations (1,000 points), the treatment with CAPD+ICO has a higher total lifetime cost compared to CAPD but provides increased QALYs. In contrast, the treatment with APD shows some instances of higher total lifetime costs than CAPD but does not yield an increase in QALYs, as observed from a societal perspective.

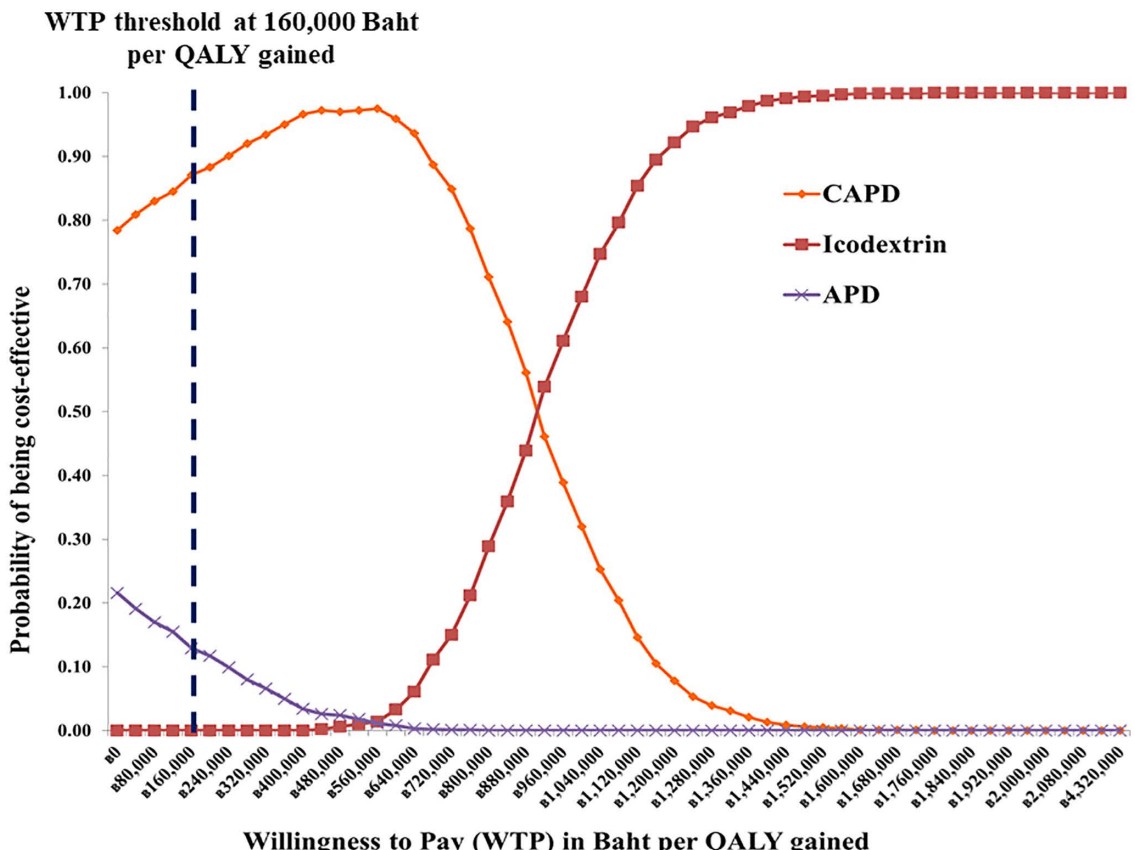

**Fig 4. Cost-effectiveness acceptability curve from the base-case analysis.**

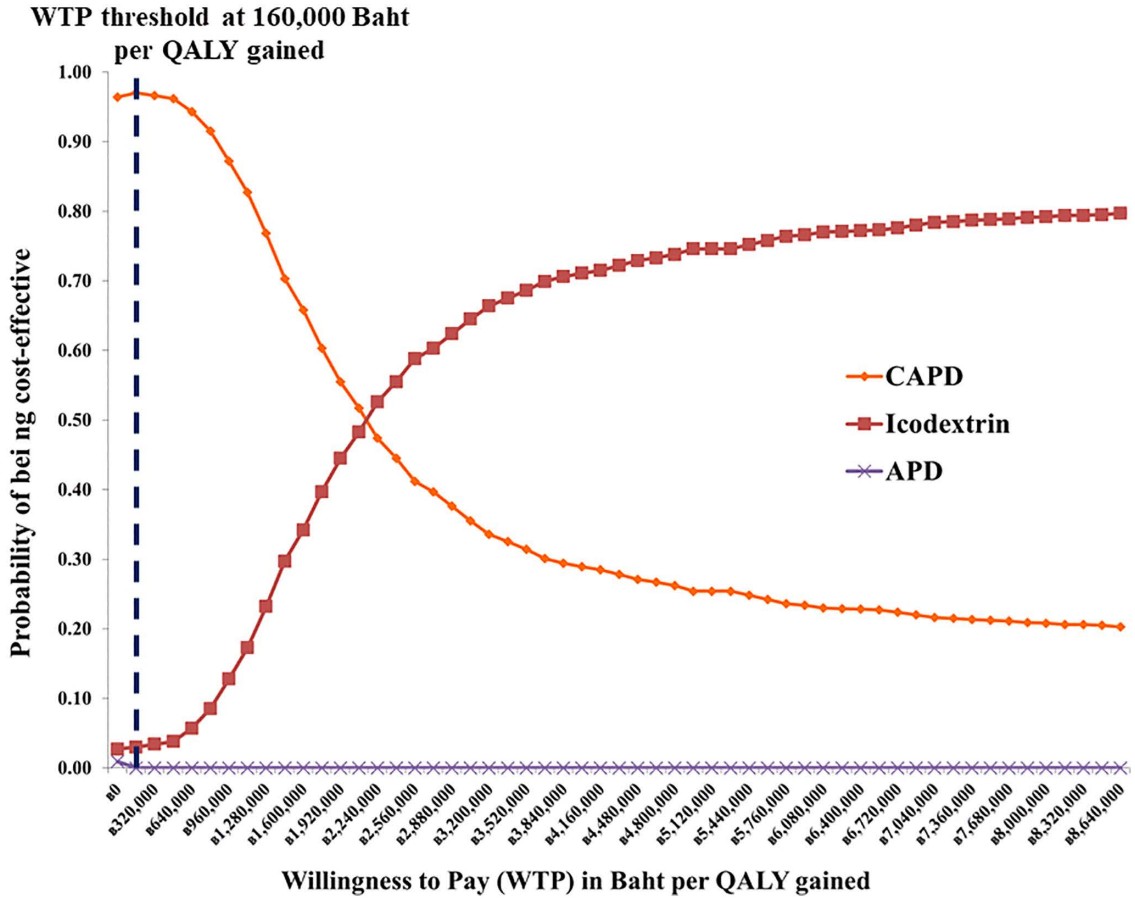

**Fig 5. Cost-effectiveness acceptability curve based on data from Yang et al.'s study in Singapore.**

## Budget impact analysis

From the study results on the budget impact, comparing CAPD with CAPD+ICO at a price of 450 THB (13 USD) per bag of icodextrin revealed that dialysis with CAPD+ICO requires an additional budget of 583 million THB (17 million USD), or an average of 58 million THB (2 million USD) per year.

## Model validation

Fig 7 illustrates the cumulative age-specific mortality rates ranging from 55–65 years for CAPD, CAPD+ICO, and APD. Overall, the modelled mortality curves show a consistent trend of increasing mortality with age, in line with the empirical data of a recent cohort study. During the earlier age intervals (55–58 years), the modelled mortality curves for APD and CAPD closely matched the real-world data, suggesting strong external validity of the modelled survival estimates of these modalities. In contrast, the CAPD+ICO strategy consistently projected lower cumulative mortality than observed in the empirical dataset, reflecting the model's assumption of enhanced survival benefits with icodextrin use.

## Discussion

Our study is the first study to conduct a cost-utility analysis of CAPD+ICO and APD compared with glucose-based CAPD using data on transitional probabilities, costs and utility values from a previous RCT study [12]. The RCT was specifically

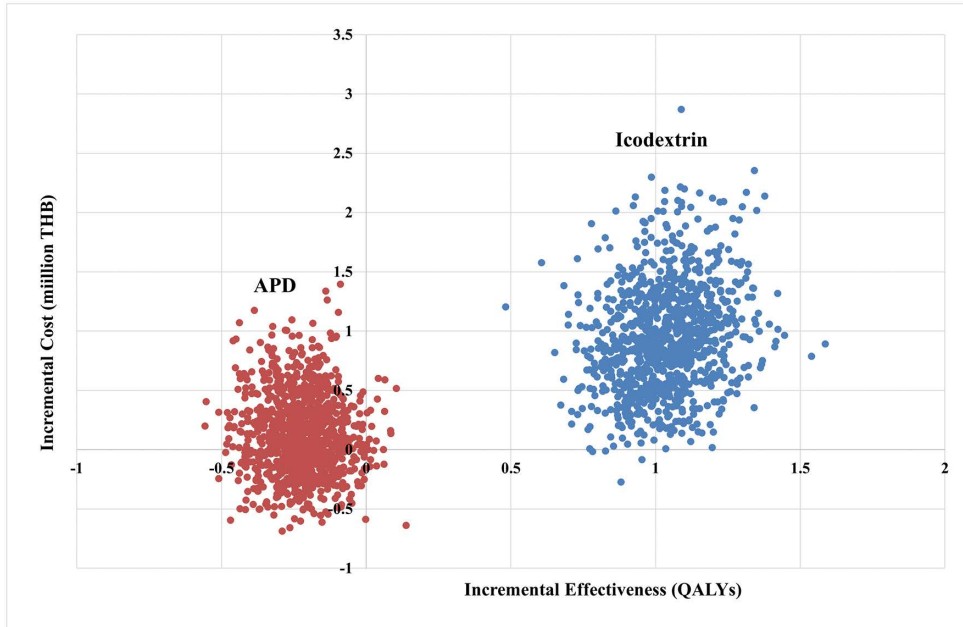

**Fig 6. Cost-effectiveness plane.**

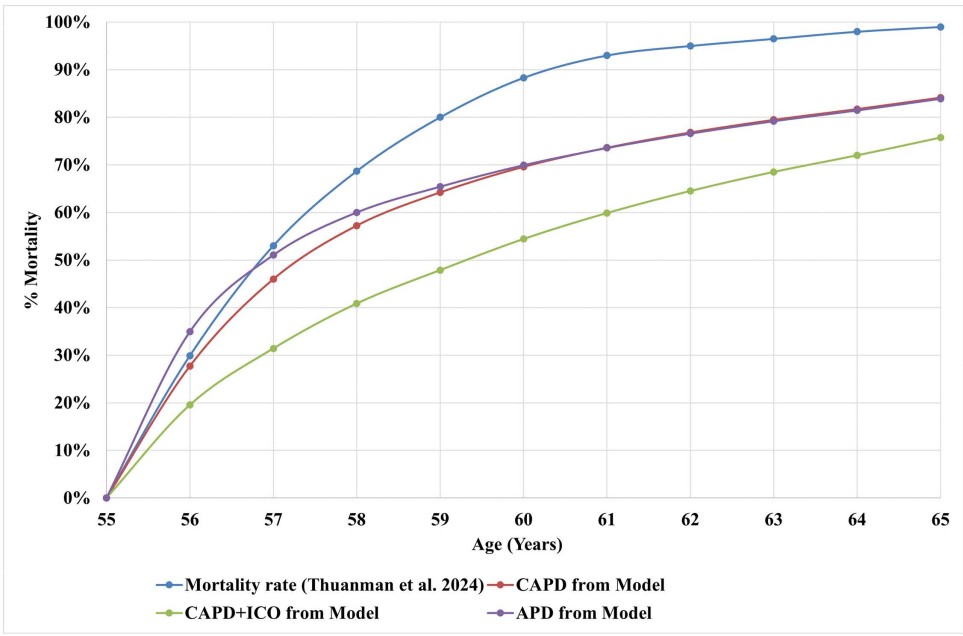

**Fig 7. Model validation results.**

designed to generate key model inputs required for this study aiming to inform policy decision making toward dialysis modality selection. When considering a cost-effectiveness threshold or WTP set at 160,000 THB (4,603 USD) per QALY gained, CAPD using glucose solution exhibited a 90% probability of being the most cost-effective option from a societal

perspective compared to both CAPD+ICO and APD. Our findings also indicate that CAPD+ICO incurred the highest lifetime cost, while providing the greatest increase in QALYs, while CAPD using glucose solution had the lowest lifetime cost but also yielded the least increase in QALYs. In addition, APD was found to be less favorable than glucose-based CAPD, as it resulted in higher lifetime costs and lower QALYs. This unfavorable outcome was primarily driven by higher medical and dialysis solution expenses associated with APD, compounded by a greater incidence of adverse events that necessitated inpatient care, which contributed to a higher mortality rate.

Moreover, patients in the APD group experienced a significantly greater decline in EQ-5D-5L scores over time compared to those in CAPD and CAPD+ICO groups, aligning with clinical findings that suggest poorer outcomes among APD patients in this context. While APD has been associated with greater convenience and improved HRQoL in high-income settings, evidence from other contexts has shown more variable results [18,19]. Similarly, a study from Singapore reported only marginal improvements in specific physical health domains among APD patients, with no significant differences in overall HRQoL compared to CAPD [16]. These findings suggest that the benefits of APD may not be universally experienced. Variability in patient characteristics, availability of caregiver support, health system infrastructure, and dialysis practices may influence patient-reported outcomes. In resource-limited settings such as Thailand, the practical advantages of APD may be diminished if patients rely heavily on family members for machine setup, monitoring, and troubleshooting. As a result, both informal caregiver costs for daily life support and formal caregiver costs were assumed to be similar between the APD and CAPD groups in our study. This dependence can also reduce the perceived convenience and potentially impact quality of life. Although the current NHSO policy promotes APD as the preferred modality, our results demonstrate its relatively reduced cost-effectiveness. These findings are consistent with previous research conducted at Siriraj Hospital, which also found that APD was associated with higher lifetime costs and lower QALYs when compared to CAPD using glucose solution [20,21]. However, the observed differences in cost and utility outcomes reflect the real-world implementation of APD in the Thai setting, where some expected advantages may not be fully realized or adequately captured. Therefore, future studies should explore on other potential benefits of APD within the Thai context.

However, our RCT study was not specifically powered to estimate long-term outcomes such as mortality and switching rates between dialysis modalities. Therefore, we conducted a scenario analysis incorporating mortality rates and EQ-5D–based utility values from a study conducted in Singapore [16]. The scenario analysis highlights that while the overall cost-effectiveness conclusions remained unchanged, the results were moderately sensitive to variations in mortality and health utility inputs. CAPD+ICO remained not cost-effective at the Thai societal WTP threshold, and APD continued to be dominated in both scenarios with even greater QALY losses under the external assumptions. However, the incremental LYs for APD increased from −0.14 in the base case to 0.18 in the scenario using Singaporean clinical and utility data. This change reflects improved survival assumptions derived from the Singaporean dataset, where patients on APD experienced better outcomes compared to those in the Thai RCT-based base case. These findings underscore the influence of data sources on model outcomes and highlight that contextual clinical effectiveness plays a role in determining LY gained and ultimately cost-effectiveness results.

The one-way sensitivity analysis results demonstrated that the price of dialysis solutions was not a significant variable, as changes in these prices did not affect the cost-effectiveness results. Even with a hypothetical reduction of icodextrin's price to zero, CAPD+ICO remained economically unviable. Furthermore, the disproportionately high inpatient cost per event in the CAPD group during the second year was primarily driven by a single outlier case at one study site, where a patient was admitted to the intensive care unit and subsequently passed away. This patient had an extended length of stay (LOS) of 63 days, which significantly increased the average cost for that group during the period. To evaluate the impact of this outlier, we conducted sensitivity analyses. Although the ICER was sensitive to variations in inpatient costs, the relative cost-effectiveness ranking of the treatment options remained unchanged, demonstrating the robustness of the model's conclusions.

In addition, the higher rate of transition to HD in the CAPD+ICO group, compared to the CAPD group, warrants consideration. All patients enrolled in the trial had fluid and sodium overload at baseline, representing a clinically complex and high-risk population [12]. Volume overload is a known predictor of adverse outcomes in PD patients, including increased risks of technique failure and mortality [22,23]. Although icodextrin is intended to improve ultrafiltration, factors such as peritoneal membrane dysfunction, persistent fluid overload, or severe infections may still necessitate modality change. The RCT found no statistically significant differences in peritonitis rates or dialysis adequacy among the three treatment arms. However, infection-related deaths were slightly lower in the CAPD+ICO group (47%) compared to CAPD (57%) and APD (56%). Notably, the CAPD group exhibited higher mortality rates in year 1 (0.1667), year 2 (0.2449), and year 3 (0.2424) than the CAPD+ICO group (0.1167, 0.1633, and 0.1212) [12]. This suggests that patients in the CAPD+ICO group, experiencing lower mortality, may have survived long enough for complications to develop, increasing the likelihood of transitioning to HD. Indeed, the CAPD+ICO group showed higher HD transfer probabilities across all years (0.0500, 0.1020, and 0.2424) compared to the CAPD group (0.0167, 0.0408, and 0.1515). In some cases, the severity of fluid overload or inadequate response to PD may have prompted clinicians to select HD as a more immediate and controlled strategy for volume management [24].

Furthermore, estimating the number of patients for the budget impact analysis is currently quite challenging due to recent policy changes. Previously, patients under UHC scheme with ESRD were required to undergo PD first. However, as of February 1, 2022, the policy now allows patients to choose either PD or HD. This shift has resulted in some PD patients planning to switch to HD. Despite this, limitations in healthcare facilities in Thailand capable of supporting HD mean that many patients wishing to transition still need to continue with PD until they can change their treatment method. Additionally, only a portion of these PD patients may benefit from CAPD+ICO or APD, leading to limited data for estimating patient numbers in this study. Various factors that could influence this situation also remain fluid moving forward. Based on the available data, it is estimated that approximately 450–900 patients annually could benefit from CAPD+ICO or APD, maintaining this estimate remaining stable over the 10-year budget impact period.

From a purely cost-effectiveness perspective, our findings suggest that CAPD+ICO is not considered cost-effective compared to CAPD using glucose solution at a WTP threshold of 160,000 THB (4,603 USD) per QALY gained. Adopting CAPD+ICO as the standard modality would require an additional annual budget of approximately 58 million THB (2 million USD). However, it is important to note that patients receiving CAPD+ICO experienced improved clinical outcomes, including longer life expectancy and better HRQoL [12]. While these outcomes extend beyond the scope of a conventional cost-utility analysis, they are highly relevant in real-world policymaking, where decision-makers must balance economic efficiency with broader health system goals.

In Thailand, benefit package decisions are not based solely on cost-effectiveness [25]. Instead, the national HTA framework explicitly incorporates multiple criteria, including ethical, equity, and social considerations, as well as budget impact and system feasibility [26]. A key historical example is the adoption of the "PD First" policy in 2008, which mandated CAPD as the first-line dialysis modality under the UHC [27]. This decision was made despite cost-effectiveness analyses at the time indicating that both PD and HD were not cost-effective compared to conservative management, with ICERs of 672,000 THB (1,933 USD) and 806,000 THB (23,189 USD) per QALY gained, respectively [28]. These values were equivalent to 2.5 and 3 times the gross domestic product (GDP) per capita, exceeding the commonly used WTP threshold of one GDP per capita [29]. The policy was instead driven by concerns about equitable access to dialysis services, particularly for rural populations. CAPD was favored because it can be self-managed at home, unlike HD, which requires access to facility-based care concentrated in urban areas [30].

Since February 2022, the scope of benefits has expanded, allowing patients to receive HD as the first method. Subsequent studies in Thailand found that HD is more expensive but provides higher QALYs than PD from a societal perspective, with ICER values for HD ranging from 1,071,319–1,339,846 THB (30,822–38,547 USD) per additional QALY (approximately 5 times the WTP) [20,21]. These past studies in patients with ESRD reflect that even treatments deemed not economically viable due to high ICER compared to the WTP in Thailand, both PD and HD have been included in the

benefit package. This consideration is based on preventing financial ruin and adhering to ethical principles that prioritize patient-centered care, since failing to receive dialysis could lead to death. Patients can engage in shared decision-making about their dialysis options with their physicians, taking into account their individual contexts.

Our study's findings must also be interpreted in light of the recent policy shift toward an APD-first approach implemented by the NHSO in 2024. Although CAPD was identified as the most cost-effective option under Thailand's WTP threshold, and APD was found to be dominated in the subgroup of patients with fluid and sodium overload, the APD-first policy reflects broader considerations beyond cost-effectiveness, including patient preference, service logistics, and workforce distribution. Importantly, the Thai HTA Guideline [14] emphasizes that policy decisions should not be based solely on cost-effectiveness but should also incorporate ethical implications, equity concerns, and social values. In this context, APD may not be cost-effective in certain subgroups but could still be justified for patients unable to perform CAPD independently or for those requiring greater flexibility to accommodate work or caregiving responsibilities. Thus, our findings remain relevant for informing refinement of the APD-first policy by identifying clinical contexts and patient subgroups where APD use is both clinically appropriate and economically reasonable, thereby supporting Thailand's goals of UHC, patient-centered care, and equitable resource allocation.

The strengths of this study included the use of data from a national RCT [12] designed specifically to inform long-term economic evaluation of CAPD+ICO and APD versus CAPD with glucose solution. Most key inputs, i.e., transition probabilities, costs, and utility values were derived directly from the trial, improving internal validity. Patient randomization was performed across the country, and prospective data collection was utilized, enhancing the accuracy and reliability of the findings while effectively reflecting the context of Thailand. The data captured actual practices in clinical settings, allowing the model to incorporate health status transitions among the three PD methods: CAPD, APD, and CAPD+ICO. Additionally, transitional probability values were derived from real-world data, supplemented by insights from nephrology associations and expert opinions from physicians when faced with study limitations.

However, this study has several limitations that must be acknowledged. Firstly, the transitional probability data used in this study were derived from RCT that span a duration of only three years, which might not capture long-term outcomes [12]. Despite this, the research team considered the data to be sufficiently reliable as it came from the RCT study which specifically focused on patients with ESRD experiencing fluid and sodium overload, which reflected the context of care in Thailand. To enhance the external validity of our model, we conducted a formal model validation by comparing the modelled age-specific mortality rates for CAPD, CAPD+ICO, and APD with empirical data from a recent cohort study of Thai PD patients who had been hospitalized due to fluid overload. The modelled survival estimates for CAPD and APD aligned well with the observed mortality rates, particularly in the earlier age intervals, supporting the robustness of our survival assumptions for these modalities. It is important to note that the patients in the cohort study who were hospitalized for fluid overload had higher mortality rates than those included in our RCT study, which may partly explain the differences observed. This highlights the need for additional real-world evidence to further refine survival assumptions in future model iterations. Overall, the validation results provide reassurance that the model is able to replicate key survival patterns seen in routine clinical practice, thus strengthening the credibility of our findings for informing dialysis policy decisions in Thailand.

Secondly, relying on international literature for data might present limitations regarding specific populations due to potential racial differences. Thirdly, the cost data for the solutions utilized in this study might be lower than current market prices, indicating that applying the study's results to other patient insurance schemes might require careful consideration of this aspect. Fourthly, according to the RCT, there was a substantial number of dropouts in the third year, primarily due to death (45%) and transition to hemodialysis (22%), which may contribute to increased uncertainty in the model parameters for that period. The reduced number of participants during the third year of follow-up may limit the robustness of the clinical and economic outcomes derived from this time frame, thereby affecting the precision of transition probabilities and cost estimates used in the model. Lastly, the long-term mortality assumptions beyond year three were based on data from

our RCT with a follow-up period limited to three years. Due to the lack of long-term data, particularly for ESRD patients with fluid and sodium overload, we assumed a constant mortality rate beyond year three, based on the average annual mortality observed during the trial period. While this assumption is commonly applied in health economic models, it may not fully capture the variability in long-term survival outcomes. Although extensive sensitivity analyses including one-way, probabilistic, scenario analyses varying the probability of death, and model validation were performed to test the robustness of this assumption, the results should be interpreted with caution. Future studies incorporating longer-term follow-up data or real-world evidence are warranted to improve the accuracy of survival projections and enhance model validity.

## Conclusions

At the Thai societal WTP threshold of 160,000 THB (4,603 USD) per QALY gained, glucose-based CAPD had a 90% probability of being the most cost-effective option from a societal perspective. CAPD+ICO is not considered cost-effective, with an estimated additional budget impact of approximately 58 million THB (2 million USD) per year. APD was found to be a dominated option, offering lower QALYs at higher costs compared to glucose-based CAPD. The results of this study can serve as economic efficiency evidence to inform policy decisions, while also taking into account clinical outcomes and equity considerations in future UHC benefit package decisions on dialysis modalities for ESRD patients with fluid and sodium overload in Thailand.

## Supporting information

**S1 Table. Baseline characteristics of study participants.**
(DOCX)

**S2 Table. Model parameters.**
(DOCX)

**S3 Table. Scenario analysis for probability of death.**
(DOCX)

**S4 Table. Scenario analysis for mortality and utility.**
(DOCX)

## Acknowledgments

The authors would like to thank all clinical experts and health economists who helped validate the model and parameters used in this study as well as the study's results. Moreover, the authors would like to acknowledge Dr. Tanita Thaweethamcharoen and Dr. Prapaporn Noparatayaporn for their inputs in this study.

## Author contributions

**Conceptualization:** Sitaporn Youngkong, Panida Yoopetch, Montarat Thavorncharoensap, Usa Chaikledkaew, Suchai Sritippayawan.

**Data curation:** Sitaporn Youngkong, Panida Yoopetch, Montira Assanatham, Usa Chaikledkaew.

**Formal analysis:** Sitaporn Youngkong, Panida Yoopetch, Montarat Thavorncharoensap, Montira Assanatham, Usa Chaikledkaew, Suchai Sritippayawan.

**Investigation:** Sitaporn Youngkong, Panida Yoopetch, Montarat Thavorncharoensap, Montira Assanatham, Usa Chaikledkaew, Suchai Sritippayawan.

**Methodology:** Sitaporn Youngkong, Panida Yoopetch, Montarat Thavorncharoensap, Montira Assanatham, Usa Chaikledkaew, Suchai Sritippayawan.

**Project administration:** Usa Chaikledkaew.

**Supervision:** Montarat Thavorncharoensap, Usa Chaikledkaew, Suchai Sritippayawan.

**Validation:** Sitaporn Youngkong, Panida Yoopetch, Montarat Thavorncharoensap, Montira Assanatham, Usa Chaikledkaew, Suchai Sritippayawan.

**Visualization:** Sitaporn Youngkong, Usa Chaikledkaew, Suchai Sritippayawan.

**Writing – original draft:** Sitaporn Youngkong, Panida Yoopetch, Montarat Thavorncharoensap, Montira Assanatham, Usa Chaikledkaew, Suchai Sritippayawan.

**Writing – review & editing:** Sitaporn Youngkong, Panida Yoopetch, Montarat Thavorncharoensap, Montira Assanatham, Usa Chaikledkaew, Suchai Sritippayawan.

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
