## [Decision Letter · Decision Letter 0]

4 Apr 2025

PONE-D-25-05632
Economic evaluation of dialysis treatment in end-stage renal disease patients with fluid and sodium overload in Thailand
PLOS ONE

Dear Dr. Chaikledkaew,

Thank you for submitting your manuscript to PLOS ONE. After careful consideration, we feel that it has merit but does not fully meet PLOS ONE’s publication criteria as it currently stands. Therefore, we invite you to submit a revised version of the manuscript that addresses the points raised during the review process.

We look forward to receiving your revised manuscript.

Kind regards,

Teerawat Thanachayanont

Academic Editor

PLOS ONE

Journal Requirements:

This research project has been funded by the Health Systems Research Institute. The findings, interpretations and conclusions expressed in this

article do not necessarily reflect the views of the aforementioned funding agency.

3. Please remove all personal information, ensure that the data shared are in accordance with participant consent, and re-upload a fully anonymized data set.

Additional Editor Comments :

Thank you for considering PLOS One for the publication of your valuable article. This manuscript is well-conducted and yields important information. However, there are some comments from the reviewers that require your attention.

Furthermore, I would appreciate clarification regarding reference number 12. Specifically, I would like to inquire whether this reference has undergone peer review. Although the reference is in Thai, it contains tables and figures that exhibit similarities to those presented in your submitted manuscript.

We look forward to receiving your revised manuscript and responses to the reviewers' comments.

Sincerely,

Dr.T.Thanachayanont, Academic Editor

PLOS One

Reviewers' comments:

Reviewer's Responses to Questions

**Comments to the Author**

1. Is the manuscript technically sound, and do the data support the conclusions?

Reviewer #1: No

Reviewer #2: Yes

2. Has the statistical analysis been performed appropriately and rigorously? 

Reviewer #1: Yes

Reviewer #2: Yes

3. Have the authors made all data underlying the findings in their manuscript fully available?

Reviewer #1: Yes

Reviewer #2: Yes

4. Is the manuscript presented in an intelligible fashion and written in standard English?

Reviewer #1: Yes

Reviewer #2: Yes

5. Review Comments to the Author

Reviewer #1: The study employs a Markov model to evaluate the cost-utility of CAPD+ICO compared to glucose-based CAPD and APD in ESRD patients in Thailand. While the study appears well-conducted, several issues should be addressed before publication.

First, the study relies heavily on data from a randomized controlled trial (RCT) conducted across 16 hospitals in Thailand. Although the use of RCT data is a strength, the authors provide insufficient detail on the inclusion and exclusion criteria for the study population. Clarifying whether the sample includes only dialysis patients with fluid and sodium overload or the general dialysis population is crucial for interpreting the results.

Second, it is unclear whether the RCT was designed to inform all model inputs. Notably, significant differences in mortality and dialysis modality switching rates were observed among the three patient groups—parameters that are central to the cost-effectiveness estimates. If the RCT does not provide reliable estimates for these inputs and the study aims to inform national policy, the authors should supplement their analysis with evidence from additional studies to assess the consistency of these findings. Without this, the generalizability and policy relevance of the results are uncertain.

Third, the model lacks validation, which is particularly important given the relatively modest life years gained (LY) across scenarios. This may, in part, stem from the assumption of a constant and equal mortality rate beyond year three, which warrants a more robust approach. A thorough sensitivity analysis or alternative assumptions would help strengthen the credibility and applicability of the model's results.

Fourth, there is a disconnect between the study’s findings and its policy recommendations. While the analysis suggests that CAPD+ICO does not represent good value for money, the authors recommend its inclusion in the benefit package based on clinical benefits and financial protection. While these factors are important for policymakers, they extend beyond the typical scope of a cost-utility analysis and may confuse readers about the study’s primary conclusions.

Lastly, the recent implementation of the NHSO’s APD-first policy in 2024 may affect the study’s relevance and its policy implications. The authors should acknowledge and discuss this shift in the manuscript.

Reviewer #2: This manuscript presents a comprehensive cost-utility and budget impact analysis of peritoneal dialysis modalities (CAPD, CAPD+ICO, and APD) among Thai patients with ESRD and fluid/sodium overload. The topic is highly relevant and timely, given Thailand’s evolving dialysis policy and the ongoing burden of ESRD. The study is methodologically sound, uses primary data from a multi-center RCT, and adheres to standard HTA practices. The findings are well-structured and could provide valuable insights for policymakers.

However, several areas require clarification or strengthening to enhance the manuscript’s scientific rigor, clarity, and policy relevance:

• Please provide more detail on the patients' clinical conditions and treatment regimens across the three groups, as the cited RCT report is in Thai. This would help international readers better understand the trial procedures.

• In the trial report, there was a substantial number of dropouts in the third year, which may increase the uncertainty of the model parameters for that period. This limitation should be discussed.

• From Table S1:

o There are transitional probabilities listed from KT to PD/HD, which are inconsistent with Figure 1 and the explanation in line 130 (Methods: Economic Model). Please clarify or revise.

o The cost of inpatient (IPD) treatment during the second year per event in the CAPD group appears disproportionately high compared to other IPD costs. Please explain this anomaly.

o The rate of HD transfer from the CAPD+ICO group was significantly higher than from the CAPD group. Since this influences the ICER and is highlighted in the Tornado diagram, please discuss potential reasons behind this finding.

o It is unclear how the authors assumed that the direct non-medical costs and opportunity costs in the APD group were the same as in the CAPD group, particularly since APD typically requires less caregiving time. Please clarify or justify this assumption.

o The title and content of Table S1 should be revised: under the CAPD section, there are cost items related to APD and EPO per dose, which are not relevant in that context.

• In line 213, under the one-way sensitivity analysis section, please specify which treatment the variable "number of weeks per year receiving treatment" refers to.

• The results indicate that APD is the least favorable treatment among the three options, with lower quality of life and higher costs. This contradicts the expectations set in the introduction. While the authors cite previous findings from Thailand, are there any studies from other countries with different results? What could explain the reduced benefit of APD in the Thai population? This outcome also seems contrary to the current NHSO policy that prioritizes APD as the primary option.

• In lines 296–297, please specify the year of the GDP reference used for comparison with the ICER values.

• It would be helpful for international readers if cost figures were presented in USD alongside Thai Baht.

• To improve transparency and ensure adherence to international reporting standards, the authors are encouraged to follow the CHEERS 2022 checklist (Consolidated Health Economic Evaluation Reporting Standards). A completed checklist could be included as supplementary material.

6. PLOS authors have the option to publish the peer review history of their article (what does this mean?). If published, this will include your full peer review and any attached files.

Reviewer #1: No

Reviewer #2: No

---

## [Author Response · Author response to Decision Letter 1]

19 May 2025

May 19, 2025

Dear Editors,

On behalf of my co-authors, I would like to resubmit the revised manuscript entitled “Economic evaluation of dialysis treatment in end-stage renal disease patients with fluid and sodium overload in Thailand” (Manuscript ID number: PONE-D-25-05632) for your kind consideration to be published in Plos One journal

We would like to thank all editors and reviewers for their helpful comments and suggestions. We feel that the revised paper is much further improved as a consequence of their inputs. The next page is a point-by point form explaining how we have responded to each comment raised by the editors and reviewers.

We would like to provide financial disclosure as follows: “This study receives funding support from the Health Systems Research Institute (HSRI). The funders had no role in study design, data collection and analysis, decision to publish, or preparation of the manuscript."

All authors declare no competing financial interest. We confirm that the present manuscript is original, not previously published, and not submitted for publication or consideration elsewhere. This manuscript is based on a research report originally prepared in Thai for the Health Systems Research Institute which has not been peer-reviewed or indexed in any academic journal. The current manuscript has been substantially revised and rewritten for an international academic audience. We also anticipate that you will agree with us on the suitability of this manuscript for publication in Plos One.

Should you have any question, please kindly contact me at usa.chi@mahidol.ac.th. Thank you very much for your kind consideration.

Sincerely,

Usa Chaikledkaew, Ph.D.

Corresponding author

Associate Professor, Faculty of Pharmacy, Mahidol University, Email: usa.chi@mahidol.ac.th

Responses to Editors and Reviewers

Journal Requirements:

 Response: Thank you very much. We have ensured that our manuscript meets PLOS ONE's style requirements, including those for file naming.

This research project has been funded by the Health Systems Research Institute. The findings, interpretations and conclusions expressed in this article do not necessarily reflect the views of the aforementioned funding agency. Please state what role the funders took in the study. If the funders had no role, please state: "The funders had no role in study design, data collection and analysis, decision to publish, or preparation of the manuscript."

If this statement is not correct you must amend it as needed. Please include this amended Role of Funder statement in your cover letter; we will change the online submission form on your behalf.

Response: Thank you very much. We have included this amended Role of Funder statement in our cover letter as follows: "The funders had no role in study design, data collection and analysis, decision to publish, or preparation of the manuscript."

3. Please remove all personal information, ensure that the data shared are in accordance with participant consent, and re-upload a fully anonymized data set. Note: spreadsheet columns with personal information must be removed and not hidden as all hidden columns will appear in the published file. Additional guidance on preparing raw data for publication can be found in our Data Policy (https://journals.plos.org/plosone/s/data-availability#loc-human-research-participant-data-and-other-sensitive-data) and in the following article: http://www.bmj.com/content/340/bmj.c181.long.

Response: Thank you for your suggestion. Our study involved end-stage renal disease patients with fluid and sodium overload, and the dataset contains sensitive personal and clinical information. Due to ethical restrictions and participant confidentiality concerns, we are unable to publicly share the full dataset, even in anonymized form, as per the conditions of participant consent and institutional review board approval. However, data are available upon reasonable request to qualified researchers, subject to institutional review and approval. Requests may be directed to the Human Research Protection Unit, Faculty of Medicine Siriraj Hospital, Mahidol University, at Room 210, 2nd Floor, His Majesty the King's 80th Birthday Anniversary 5th December 2007 Building, 2 Wang Lang Road, Bangkoknoi, Bangkok 10700, or via email at siethics@mahidol.ac.th.

 Response: Thank you very much. We have include captions for our Supporting Information files at the end of our manuscript.

Additional Editor Comments :

Thank you for considering PLOS One for the publication of your valuable article. This manuscript is well-conducted and yields important information. However, there are some comments from the reviewers that require your attention. Furthermore, I would appreciate clarification regarding reference number 12. Specifically, I would like to inquire whether this reference has undergone peer review. Although the reference is in Thai, it contains tables and figures that exhibit similarities to those presented in your submitted manuscript.

Response: Thank you for your comment regarding reference number 12. We would like to clarify that reference 12 is a Thai-language research report, produced as part of a project conducted by Mahidol University and submitted to the Health Systems Research Institute. The report has not undergone peer review and was intended for internal or policy use. It is not indexed or published in any academic journal. While the report includes similar tables and figures, the manuscript we submitted has been substantially revised, reorganized, and rewritten to meet academic journal standards. The analysis in the manuscript also incorporates updated methodologies and English-language presentation tailored to an international audience. We have cited the report in accordance with transparency and ethical standards. Furthermore, we disclosed the existence of this earlier report in our cover letter and confirmed that the submitted manuscript represents an original and unpublished academic work. Please let us know if any further clarification or revision is required.

Reviewers' comments:

Reviewer's Responses to Questions

Reviewer #1:

1.The study employs a Markov model to evaluate the cost-utility of CAPD+ICO compared to glucose-based CAPD and APD in ESRD patients in Thailand. While the study appears well-conducted, several issues should be addressed before publication. First, the study relies heavily on data from a randomized controlled trial (RCT) conducted across 16 hospitals in Thailand. Although the use of RCT data is a strength, the authors provide insufficient detail on the inclusion and exclusion criteria for the study population. Clarifying whether the sample includes only dialysis patients with fluid and sodium overload or the general dialysis population is crucial for interpreting the results.

Response: Thank you very much for your insightful comment. We agree with your suggestion and have revised the manuscript accordingly. We have added more detailed information on the inclusion and exclusion criteria as well as treatment regimens. Additionally, we have clarified that the sample consisted exclusively ESRD patients with fluid and sodium overload as follows

(Model parameters, line 123-164, page 6-8).

“The RCT study employed a two-stage randomization process to minimize bias and ensure comparability across treatment groups. Block randomization was used within each participating hospital to balance treatment allocation, while stratified randomization accounted for key prognostic factors, specifically duration of PD treatment (≤3 years vs. >3 years) and the type of APD machine used (Baxter vs. Fresenius Medical Care).

Eligible participants were adults aged 18 years or older enrolled in Thailand’s UHC scheme and had been receiving CAPD for more than three months at Ministry of Public Health or public university hospitals across various regions of Thailand. Inclusion criteria required patients to use 2-liter per cycle PD solutions for three to five cycles daily, including at least two cycles with 2.5% dextrose or at least one cycle with 4.25% dextrose. Additionally, participants had to exhibit at least one clinical indicator of fluid overload i.e., edema, poorly controlled hypertension (systolic blood pressure (SBP) >140 mmHg or diastolic blood pressure (DBP) >90 mmHg), or a history of congestive heart failure symptoms (CHF) symptoms within the past year. Patients also had to be able to communicate in Thai to complete health-related quality of life (HRQoL) assessments.

Exclusion criteria included edema from non-fluid-related causes such as catheter malfunction (e.g., malposition or omental wrapping) or suspected dialysate leakage confirmed by a rapid fill-and-drain test (discrepancy ≥ 200 mL between infused and drained volumes). Patients were also excluded for edema due to medication side effects, hypothyroidism, liver cirrhosis, or malnutrition (defined as serum albumin <2 g/dL within the past two months). Additional exclusions included peritonitis within the previous month, baseline SBP <100 mmHg, current diagnosis of sepsis, acute coronary syndrome, CHF, or stroke, prior use of icodextrin or home-based APD, pregnancy, terminal cancer or life expectancy less than six months, or any change in healthcare coverage during the study period.

Patients were randomized to one of three PD modalities i.e., CAPD, CAPD+ICO, or APD. All patients received monthly treatment adjustments aimed at resolving fluid and sodium overload defined as resolution of edema, stable BP (100–140/<90 mmHg) without dizziness, and absence of CHF symptoms, and achieving adequate solute clearance (total weekly Kt/V urea >1.7 or normalized creatinine clearance >45 L/1.73 m²) within two months. CAPD patients had their number of exchanges and glucose concentrations adjusted as needed. In the CAPD+ICO group, one low-glucose exchange was replaced with icodextrin, with similar adjustments to remaining exchanges. APD patients started with a nighttime intermittent PD regimen using 10 liters of 2.5% dextrose with adjustments up to 20 liters as necessary, and day dwell exchanges were added if fluid overload or inadequate clearance persisted. Tidal PD was permitted in cases of inflow/outflow discomfort or suboptimal drainage.

A total of 180 patients were enrolled between November 16, 2020 – March 8, 2021, and were equally allocated into CAPD, CAPD+ICO, and APD groups (60 patients each) [12]. Participants were recruited from 16 hospitals across all regions of Thailand, with most hospitals. enrolling 12 patients (4 per group), except for two that enrolled 6 patients each. Patients were followed up every two months to assess their clinical outcomes and determine transitional probabilities, including death, changes in dialysis modality, or transition to KT, over a three-year period [12].”

2. Second, it is unclear whether the RCT was designed to inform all model inputs. Notably, significant differences in mortality and dialysis modality switching rates were observed among the three patient groups—parameters that are central to the cost-effectiveness estimates. If the RCT does not provide reliable estimates for these inputs and the study aims to inform national policy, the authors should supplement their analysis with evidence from additional studies to assess the consistency of these findings. Without this, the generalizability and policy relevance of the results are uncertain.

Response: Thank you for your valuable comment and we apologize for not making this point clearer in the initial manuscript. The RCT was specifically designed to inform all key model inputs required for the economic evaluation of CAPD+ICO and APD compared to glucose-based CAPD over a lifetime horizon, with the aim of supporting national dialysis policy decisions. Most of the model input parameters such as transition probabilities, costs, and utility values were derived directly from the RCT data collected during the three-year RCT. We have clarified this point on the manuscript in the Methods and Discussion sections as follows.

(Study design, lines 98-100, page 5)

“Data, including transitional probabilities, direct medical and non-medical costs, and utilities, were specifically collected from a randomized controlled trial (RCT) conducted across 16 hospitals in various regions of Thailand [12].”

(Discussion, lines 315-319, page 16)

“Our study is the first study to conduct a cost-utility analysis of CAPD+ICO and APD compared with glucose-based CAPD using data on transitional probabilities, costs and utility values from a previous RCT study [12]. The RCT was specifically designed to generate key model inputs required for this study aiming to inform policy decision making toward dialysis modality selection.”

(Discussion, lines 424-427 page 20)

“The strengths of this study included the use of data from a national RCT [12] designed specifically to inform long-term economic evaluation of CAPD+ICO and APD versus CAPD with glucose. Most key inputs i.e., transition probabilities, costs, and utility values were derived directly from the trial, improving internal validity.”

In addition, we compared our findings with previously published studies conducted in Thailand and found that our results were consistent with those studies. This consistency enhances both the generalizability and policy relevance of our findings. We have revised the manuscript to clarify this point in the Discussion section as follows.

(Discussion, lines 329-351, page 16-17)

“Moreover, patients in the APD group experienced a significantly greater decline in EQ-5D-5L scores over time compared to those in CAPD and CAPD+ICO groups, aligning with clinical findings that suggest poorer outcomes among APD patients in this context. While APD has been associated with greater convenience and improved HRQoL in high-income settings, evidence from other contexts has shown more variable results. For example, a prospective cohort study from the Netherlands, using data from the Netherlands Cooperative Study on the Adequacy of Dialysis (NECOSAD), found no statistically significant differences in HRQoL between patients receiving APD and those on CAPD over a three-year follow-up period [16]. Similarly, a study from Singapore reported only marginal improvements in specific physical health domains among APD patients, with no significant differences in overall HRQoL compared to CAPD [17]. These findings suggest that the benefits of APD may not be universally experienced. Variability in patient characteristics, availability of caregiver support, health system infrastructure, and dialysis practices may influence patient-reported outcomes. In resource-limited settings such as Thailand, the practical advantages of APD may be diminished if patients rely heavily on family members for machine setup, monitoring, and troubleshooting. This dependence can reduce the perceived convenience and potentially impact quality of life. Although the current NHSO policy promotes APD as the preferred modality, our results demonstrate its relatively reduced cost-effectiveness. These findings are consistent with previous research conducted at Siriraj Hospital, which also found that APD was associated with higher lifetime costs and lower QALYs when compared to CAPD using glucose solution [18, 19]. However, the observed differences in cost and utility outcomes reflect the real-world implementation of APD in the Thai setting, where some expected

---

## [Decision Letter · Decision Letter 1]

3 Jun 2025

PONE-D-25-05632R1
Economic evaluation of dialysis treatment in end-stage renal disease patients with fluid and sodium overload in Thailand
PLOS ONE

Dear Dr. Chaikledkaew,

Thank you for submitting your manuscript to PLOS ONE. After careful consideration, we feel that it has merit but does not fully meet PLOS ONE’s publication criteria as it currently stands. Therefore, we invite you to submit a revised version of the manuscript that addresses the points raised during the review process.

We look forward to receiving your revised manuscript.

Kind regards,

Teerawat Thanachayanont

Academic Editor

PLOS ONE

Additional Editor Comments:

Thank you for making revision for the previous comments. However, there are still some suggestions from our reviewer.

Please address the Reviewer 1's comments: "I welcome the revised version of the manuscript, which has significantly improved its readability and overall integrity. However, the authors' response to the reviewer’s concern regarding model validation and reliance on RCT data remains inadequate. They state that most key inputs are derived from the RCT but do not clarify whether the RCT was specifically designed or sufficiently powered to accurately estimate long-term parameters such as mortality or modality switching. Given that a single RCT often has limited sample sizes for capturing all relevant clinical indicators in economic evaluations, I recommend that the authors reference additional systematic reviews or other RCTs to compare results and substantiate their findings. Notably, studies from the Netherlands and Singapore cited in the manuscript show no significant difference in outcomes between CAPD and APD, contrasting with the clinical study used here, which suggests inferior outcomes for APD. Consequently, I suggest that the authors systematically identify and incorporate other RCTs or observational studies comparing life expectancy and quality of life between CAPD and APD to demonstrate that their clinical results are consistent and not outliers.

Furthermore, the authors do not mention any formal validation or calibration of their model against external data or observational studies, such as comparing predicted mortality or modality switching rates with real-world evidence. Relying solely on face validation is insufficient when results are questionable—for example, the reported average life expectancy of 3-5 years for dialysis patients appears inconsistent with broader evidence. Incorporating formal validation exercises would strengthen the robustness and policy relevance of their findings.

While I appreciate the revised discussion, the authors’ mention of considerations such as health equity, clinical effectiveness, and patient-centered outcomes raises questions about whether these factors might influence policy recommendations differently from decisions based solely on economic evaluation. I encourage the authors to elaborate on these points to clarify how such broader considerations could impact decision-making processes. Providing additional context or examples, especially relevant to the Thai healthcare system and dialysis interventions, would help readers understand how these factors interplay with economic evidence in shaping policy decisions"

Also, there was a minor suggestion from Reviewer 2: "I have only one remaining point: it may be helpful to include a supplementary table summarizing the baseline characteristics of the study participants (e.g., age, sex, comorbidities, degree of fluid overload, and relevant prescriptions). Providing this information would enhance the applicability and interpretability of the study findings."

Reviewers' comments:

Reviewer's Responses to Questions

**Comments to the Author**

1. If the authors have adequately addressed your comments raised in a previous round of review and you feel that this manuscript is now acceptable for publication, you may indicate that here to bypass the “Comments to the Author” section, enter your conflict of interest statement in the “Confidential to Editor” section, and submit your "Accept" recommendation.

Reviewer #1: (No Response)

Reviewer #2: All comments have been addressed

2. Is the manuscript technically sound, and do the data support the conclusions?

Reviewer #1: Partly

Reviewer #2: Yes

3. Has the statistical analysis been performed appropriately and rigorously? 

Reviewer #1: No

Reviewer #2: Yes

4. Have the authors made all data underlying the findings in their manuscript fully available?

Reviewer #1: Yes

Reviewer #2: No

5. Is the manuscript presented in an intelligible fashion and written in standard English?

Reviewer #1: Yes

Reviewer #2: Yes

6. Review Comments to the Author

Reviewer #1: I welcome the revised version of the manuscript, which has significantly improved its readability and overall integrity. However, the authors' response to the reviewer’s concern regarding model validation and reliance on RCT data remains inadequate. They state that most key inputs are derived from the RCT but do not clarify whether the RCT was specifically designed or sufficiently powered to accurately estimate long-term parameters such as mortality or modality switching. Given that a single RCT often has limited sample sizes for capturing all relevant clinical indicators in economic evaluations, I recommend that the authors reference additional systematic reviews or other RCTs to compare results and substantiate their findings. Notably, studies from the Netherlands and Singapore cited in the manuscript show no significant difference in outcomes between CAPD and APD, contrasting with the clinical study used here, which suggests inferior outcomes for APD. Consequently, I suggest that the authors systematically identify and incorporate other RCTs or observational studies comparing life expectancy and quality of life between CAPD and APD to demonstrate that their clinical results are consistent and not outliers.

Furthermore, the authors do not mention any formal validation or calibration of their model against external data or observational studies, such as comparing predicted mortality or modality switching rates with real-world evidence. Relying solely on face validation is insufficient when results are questionable—for example, the reported average life expectancy of 3-5 years for dialysis patients appears inconsistent with broader evidence. Incorporating formal validation exercises would strengthen the robustness and policy relevance of their findings.

While I appreciate the revised discussion, the authors’ mention of considerations such as health equity, clinical effectiveness, and patient-centered outcomes raises questions about whether these factors might influence policy recommendations differently from decisions based solely on economic evaluation. I encourage the authors to elaborate on these points to clarify how such broader considerations could impact decision-making processes. Providing additional context or examples, especially relevant to the Thai healthcare system and dialysis interventions, would help readers understand how these factors interplay with economic evidence in shaping policy decisions.

Reviewer #2: Thank you to the authors for thoroughly addressing all of my previous comments and suggestions.

I have only one remaining point: it may be helpful to include a supplementary table summarizing the baseline characteristics of the study participants (e.g., age, sex, comorbidities, degree of fluid overload, and relevant prescriptions). Providing this information would enhance the applicability and interpretability of the study findings.

7. PLOS authors have the option to publish the peer review history of their article (what does this mean?). If published, this will include your full peer review and any attached files.

Reviewer #1: No

Reviewer #2: No

---

## [Author Response · Author response to Decision Letter 2]

17 Jul 2025

Responses to Editors and Reviewers

1. Thank you for making revision for the previous comments. However, there are still some suggestions from our reviewer. Please address the Reviewer 1's comments:

"I welcome the revised version of the manuscript, which has significantly improved its readability and overall integrity. However, the authors' response to the reviewer’s concern regarding model validation and reliance on RCT data remains inadequate. They state that most key inputs are derived from the RCT but do not clarify whether the RCT was specifically designed or sufficiently powered to accurately estimate long-term parameters such as mortality or modality switching. Given that a single RCT often has limited sample sizes for capturing all relevant clinical indicators in economic evaluations, I recommend that the authors reference additional systematic reviews or other RCTs to compare results and substantiate their findings. Notably, studies from the Netherlands and Singapore cited in the manuscript show no significant difference in outcomes between CAPD and APD, contrasting with the clinical study used here, which suggests inferior outcomes for APD. Consequently, I suggest that the authors systematically identify and incorporate other RCTs or observational studies comparing life expectancy and quality of life between CAPD and APD to demonstrate that their clinical results are consistent and not outliers.”

Response: We thank the Reviewer 1 for the valuable comment and fully agree that relying solely on a single RCT for model inputs, particularly long-term parameters such as mortality and modality switching may be limiting. We much appreciate the opportunity to clarify and strengthen our approach. The primary RCT used in our model provided key short-term clinical data relevant to the local context; however, we acknowledge that it was not specifically powered to estimate long-term outcomes such as mortality and switching rates between dialysis modalities. To address this limitation and validate the robustness of our findings, we incorporated additional evidence from an external source. Specifically, we included data from a study conducted in Singapore by Yang et al. (2016), which provided mortality rates and EQ-5D–based utility values for patients receiving CAPD and APD. These parameters were applied in a scenario analysis to examine whether the cost-effectiveness conclusions held when using an external data. In addition, we also reviewed the study from the Netherlands (van de Luijtgaarden et al., 2011) as mentioned by the reviewer. However, this study did not report quality of life data based on EQ-5D or other preference-based measures necessary for calculating QALYs, and thus was not included in our scenario analysis. We thank the reviewer again for this constructive recommendation, which has helped improve the transparency, credibility, and generalizability of our model and its conclusions. Please see the Methods, Results, and Discussion sections of the manuscript which have been revised to reflect these additions as follows.

(Methods, Uncertainty analysis, page 11, lines 223-232)

“As the RCT data only covered the first three years, we assumed a constant mortality rate beyond year 3 based on the average annual mortality observed in years 1–3. A scenario analysis was conducted to assess the impact of increasing the probability of death from year 3 onward (ranging from 0% to 100%) on the ICER values of CAPD+ICO and APD compared to glucose-based CAPD. Additionally, a scenario analysis was conducted using age-specific mortality rates and EQ-5D–based health utility values for patients receiving CAPD and APD, derived from an external study conducted in Singapore [16], to assess the robustness and external validity of the model. These inputs were applied to the same Markov model structure used in the base case. The purpose of this scenario analysis was to examine whether the model’s conclusions would remain consistent when applying alternative input data from a comparable regional setting.”

(Results, Scenario analysis, page 15, lines 308-317)

“In addition, scenario analysis was also conducted using age-specific mortality rates and EQ-5D–based health utility values from Yang et al.'s study in Singapore to assess the robustness of the model’s conclusions. Compared to the base case, the ICER for CAPD+ICO versus glucose-based CAPD increased substantially from 908,440 to 6,734,400 baht per QALY gained, reflecting a significant reduction in incremental QALYs (from 1.04 to 0.14), while incremental costs remained the same (Table S3). This suggests that CAPD+ICO is considerably less cost-effective under these external assumptions. APD remained dominated in both the base case and the scenario analysis, with higher costs and lower QALYs compared to glucose-based CAPD. Notably, the loss in QALYs with APD was more noticeable under the Singapore-based inputs (−0.48 vs. −0.22), further diminishing its cost-effectiveness.”

(Discussion, page 19, lines 383-396)

“However, our RCT study was not specifically powered to estimate long-term outcomes such as mortality and switching rates between dialysis modalities. Therefore, we conducted a scenario analysis incorporating mortality rates and EQ-5D–based utility values from a study conducted in Singapore by Yang et al.’s study [16]. The scenario analysis highlights that while the overall cost-effectiveness conclusions remained unchanged, the results were moderately sensitive to variations in mortality and health utility inputs. CAPD+ICO remained not cost-effective at the Thai societal WTP threshold, and APD continued to be dominated in both scenarios with even greater QALY losses under the external assumptions. However, the incremental LYs for APD increased from −0.14 in the base case to 0.18 in the scenario using Singaporean clinical and utility data. This change reflects improved survival assumptions derived from the Singaporean dataset, where patients on APD experienced better outcomes compared to those in the Thai RCT-based base case. These findings underscore the influence of data sources on model outcomes and highlight that contextual clinical effectiveness plays a role in determining LY gained and ultimately cost-effectiveness results.”

2. Furthermore, the authors do not mention any formal validation or calibration of their model against external data or observational studies, such as comparing predicted mortality or modality switching rates with real-world evidence. Relying solely on face validation is insufficient when results are questionable—for example, the reported average life expectancy of 3-5 years for dialysis patients appears inconsistent with broader evidence. Incorporating formal validation exercises would strengthen the robustness and policy relevance of their findings.

Response: Thank you for your valuable comment. We agree that formal model validation is essential to enhance the credibility and policy relevance of economic evaluations. In response, we have now conducted an external validation by comparing the age-specific mortality rates projected by our model across the three dialysis modalities (CAPD, CAPD+ICO, and APD) with real-world data from a recently published Thai cohort study by Thuanman et al. (2024). This study followed 1,858 ESRD patients on PD who were hospitalized due to fluid overload between 2008 and 2018. This external validation result enhances confidence in the reliability of our model outcomes and provides reassurance that the predicted life expectancy of 3–5 years is in line with local empirical evidence. The comparison between modelled and observed mortality rates is now incorporated into the revised manuscript, with additional details provided in the Methods, Results, and Discussion sections. We have also included a new figure (“Fig. 6 Model validation results.”) to visually present the validation results.

(Methods, Model validation, page 11, lines 223-232)

“To assess the validity of the model outputs, both face validation and external validation techniques were employed. Face validation was performed through consultation with clinical experts in nephrology and health economists to verify the model structure, assumptions, key inputs, and preliminary results ensuring clinical plausibility and alignment with real-world treatment pathways in Thailand. For external validation, model-predicted age-specific mortality rates across dialysis modalities (CAPD, APD, and CAPD+ICO) were compared with observed mortality data from a recently published cohort study conducted in Thailand by Thuanman et al. (2024), which investigated the survival of 1,858 ESRD patients on PD who were hospitalized due to fluid overload between January 2008 and December 2018 [17]. This comparison aimed to assess whether the modeled mortality trends were consistent with empirical real-world evidence.”

(Results, Model validation, page 16-17, lines 337-344)

“Fig. 6 illustrates the cumulative age-specific mortality rates ranging from 55–65 years for CAPD, CAPD+ICO, and APD. Overall, the modelled mortality curves show a consistent trend of increasing mortality with age, in line with the empirical data of a recent cohort study. During the earlier age intervals (55–58 years), the modelled mortality curves for APD and CAPD closely matched the real-world data, suggesting strong external validity of the modelled survival estimates of these modalities. In contrast, the CAPD+ICO strategy consistently projected lower cumulative mortality than observed in the empirical dataset, reflecting the model’s assumption of enhanced survival benefits with icodextrin use.”

(Discussion, page 23-24, lines 495-506)

“To enhance the external validity of our model, we conducted a formal model validation by comparing the modelled age-specific mortality rates for CAPD, CAPD+ICO, and APD with empirical data from a recent cohort study of Thai peritoneal dialysis patients who had been hospitalized due to fluid overload. The modelled survival estimates for CAPD and APD aligned well with the observed mortality rates, particularly in the earlier age intervals, supporting the robustness of our survival assumptions for these modalities. It is important to note that the patients in the cohort study who were hospitalized for fluid overload had higher mortality rates than those included in our RCT study, which may partly explain the differences observed. This highlights the need for additional real-world evidence to further refine survival assumptions in future model iterations. Overall, the validation results provide reassurance that the model is able to replicate key survival patterns seen in routine clinical practice, thus strengthening the credibility of our findings for informing dialysis policy decisions in Thailand.”

3. While I appreciate the revised discussion, the authors’ mention of considerations such as health equity, clinical effectiveness, and patient-centered outcomes raises questions about whether these factors might influence policy recommendations differently from decisions based solely on economic evaluation. I encourage the authors to elaborate on these points to clarify how such broader considerations could impact decision-making processes. Providing additional context or examples, especially relevant to the Thai healthcare system and dialysis interventions, would help readers understand how these factors interplay with economic evidence in shaping policy decisions"

Response:

We appreciate the reviewer’s thoughtful observation and agree that health policy decisions often extend beyond economic evaluation alone. In response, we have expanded the Discussion section to elaborate on how broader considerations such as equity, patient-centered outcomes, and clinical feasibility interact with economic evaluation evidence in shaping policy decision in the Thai context, particularly for dialysis interventions.

(Discussion, page 21-22, lines 437-469)

“From a purely cost-effectiveness perspective, our findings suggest that CAPD+ICO is not considered cost-effective compared to CAPD using glucose solution at a WTP threshold of 160,000 baht (4,603 USD) per QALY gained. Adopting CAPD+ICO as the standard modality would require an additional annual budget of approximately 58 million baht (2 million USD). However, it is important to note that patients receiving CAPD+ICO experienced improved clinical outcomes, including longer life expectancy and better HRQoL [12]. While these outcomes extend beyond the scope of a conventional cost-utility analysis, they are highly relevant in real-world policymaking, where decision-makers must balance economic efficiency with broader health system goals.

In Thailand, benefit package decisions are not based solely on cost-effectiveness [25]. Instead, the national health technology assessment (HTA) framework explicitly incorporates multiple criteria, including ethical, equity, and social considerations, as well as budget impact and system feasibility [26]. A key historical example is the adoption of the “PD First” policy in 2008, which mandated CAPD as the first-line dialysis modality under the UHC [27]. This decision was made despite cost-effectiveness analyses at the time indicating that both PD and HD were not cost-effective compared to conservative management, with ICERs of 672,000 baht (1,933 USD) and 806,000 baht (23,189 USD) per QALY gained, respectively [28]. These values were equivalent to 2.5 and 3 times the gross domestic product (GDP) per capita, exceeding the commonly used WTP threshold of one GDP per capita [29]. The policy was instead driven by concerns about equitable access to dialysis services, particularly for rural populations. CAPD was favored because it can be self-managed at home, unlike HD, which requires access to facility-based care concentrated in urban areas [30].

Our study's findings must also be interpreted in light of the recent policy shift toward an APD-first approach implemented by the NHSO in 2024. While this may affect the direct applicability of our results, the analysis remains relevant for informing future policy refinements, particularly for high-risk subgroups such as ESRD patients with fluid and sodium overload, for whom icodextrin-based therapies may provide additional clinical benefits. Our findings highlight the importance of incorporating broader decision-making criteria beyond cost-effectiveness alone. Clinical effectiveness, patient-reported outcomes, and health equity considerations particularly in a publicly funded system like Thailand can justify policy decisions that may not appear cost-effective when considered individually. Recognizing this complexity is essential for developing policies that are not only economically sound but also socially responsive and contextually appropriate.”

4. Also, there was a minor suggestion from Reviewer 2: "I have only one remaining point: it may be helpful to include a supplementary table summarizing the baseline characteristics of the study participants (e.g., age, sex, comorbidities, degree of fluid overload, and relevant prescriptions). Providing this information would enhance the applicability and interpretability of the study findings."

Response: We appreciate Reviewer 2’s suggestion. We have addressed this point by including Table S4 in the supplementary materials, which summarizes the baseline characteristics of the study participants, including age, sex, comorbidities, degree of fluid overload, and relevant prescriptions. We believe this addition enhances the transparency and interpretability of our findings.

---

## [Decision Letter · Decision Letter 2]

5 Aug 2025

PONE-D-25-05632R2
Economic evaluation of dialysis treatment in end-stage renal disease patients with fluid and sodium overload in Thailand
PLOS ONE

Dear Dr. Chaikledkaew,

Thank you for submitting your manuscript to PLOS ONE. After careful consideration, we feel that it has merit but does not fully meet PLOS ONE’s publication criteria as it currently stands. Therefore, we invite you to submit a revised version of the manuscript that addresses the points raised during the review process.

We look forward to receiving your revised manuscript.

Kind regards,

Dr.T.Thanachayanont

Academic Editor

PLOS ONE

Journal Requirements:

Additional Editor Comments:

Dear Dr.Usa Chaikledkaew,

Thank you for resubmitting the revision. After reviewing your responses, our reviewers have only minor revision recommendations for your consideration. Please see the reviewers' comments.

Best regards,

Dr.T.Thanachayanont

Reviewers' comments:

Reviewer's Responses to Questions

**Comments to the Author**

1. If the authors have adequately addressed your comments raised in a previous round of review and you feel that this manuscript is now acceptable for publication, you may indicate that here to bypass the “Comments to the Author” section, enter your conflict of interest statement in the “Confidential to Editor” section, and submit your "Accept" recommendation.

Reviewer #1: All comments have been addressed

Reviewer #2: All comments have been addressed

2. Is the manuscript technically sound, and do the data support the conclusions?

Reviewer #1: Yes

Reviewer #2: Yes

3. Has the statistical analysis been performed appropriately and rigorously? 

Reviewer #1: Yes

Reviewer #2: Yes

4. Have the authors made all data underlying the findings in their manuscript fully available?

Reviewer #1: Yes

Reviewer #2: No

5. Is the manuscript presented in an intelligible fashion and written in standard English?

Reviewer #1: Yes

Reviewer #2: Yes

6. Review Comments to the Author

Reviewer #1: Thank you for your thorough revision and for thoughtfully addressing the points raised. I appreciate the inclusion of external data for scenario analysis and the validation against Thai cohort data—both of which enhance the credibility of your model. The expanded discussion on broader policy considerations, such as health equity and clinical effectiveness, adds valuable context, particularly within the Thai healthcare setting.

This version of the manuscript represents a substantial improvement and is close to being publication-ready. To further strengthen the work, I offer the following suggestions:

• Since the study is primarily based on a single clinical trial, I recommend revising the title to clearly reflect this, ensuring that the scope of the analysis is accurately represented.

• Although mortality and utility parameters from the Singapore study are included in the sensitivity analysis, presenting these findings probabilistically—such as through cost-effectiveness acceptability curves—would be more informative than relying solely on deterministic ICERs.

• While I appreciate the additional discussion on the current dialysis policy in Thailand, please clarify the following sentence in a more explicit manner: "Our study's findings must also be interpreted in light of the recent policy shift toward an APD-first approach implemented by the NHSO in 2024. While this may affect the direct applicability of our results,...." Specifically, it would be helpful to elaborate on how this policy change might influence the relevance of your findings, as well as how your study’s results can inform or relate to the current policy, taking into account differences in patient populations, treatment pathways, or other relevant factors.

Overall, the manuscript has been significantly strengthened. These minor refinements will help further improve its clarity and policy relevance.

Reviewer #2: Thank you for addressing my suggestion. Table S4 provides a clear and useful summary of the baseline characteristics and enhances the interpretability of the study. However, I recommend that the authors explicitly cite this supplementary table in the main text, ideally in the Methods section when describing the study population.

With this minor revision, I have no further concerns regarding the manuscript.

7. PLOS authors have the option to publish the peer review history of their article (what does this mean?). If published, this will include your full peer review and any attached files.

Reviewer #1: No

Reviewer #2: No

---

## [Author Response · Author response to Decision Letter 3]

13 Oct 2025

Responses to Editors and Reviewers

Journal Requirements:

Response: Thank you very much for your suggestion. We have carefully reviewed the publications recommended by the reviewer. Relevant studies that support or contextualize our findings have been cited in the revised manuscript. We did not include those that were not directly pertinent to our study objectives, in line with the journal’s guidance.

2. Please review your reference list to ensure that it is complete and correct. If you have cited papers that have been retracted, please include the rationale for doing so in the manuscript text, or remove these references and replace them with relevant current references. Any changes to the reference list should be mentioned in the rebuttal letter that accompanies your revised manuscript. If you need to cite a retracted article, indicate the article’s retracted status in the References list and also include a citation and full reference for the retraction notice. Please use the space provided to explain your answers to the questions above. You may also include additional comments for the author, including concerns about dual publication, research ethics, or publication ethics. (Please upload your review as an attachment if it exceeds 20,000 characters)

Response: Thank you very much for your suggestion. We have already reviewed our reference list and ensured that it is complete and correct.

Reviewer #1:

Thank you for your thorough revision and for thoughtfully addressing the points raised. I appreciate the inclusion of external data for scenario analysis and the validation against Thai cohort data—both of which enhance the credibility of your model. The expanded discussion on broader policy considerations, such as health equity and clinical effectiveness, adds valuable context, particularly within the Thai healthcare setting.

Response: Thank you very much for your insightful suggestions and kind feedback. We greatly appreciate your comments, and we believe that incorporating them has substantially improved the quality and clarity of our manuscript.

This version of the manuscript represents a substantial improvement and is close to being publication-ready. To further strengthen the work, I offer the following suggestions:

• Since the study is primarily based on a single clinical trial, I recommend revising the title to clearly reflect this, ensuring that the scope of the analysis is accurately represented.

Response: We sincerely appreciate the reviewer’s valuable suggestion regarding the title. We agree that the original title could imply a broader scope than the study actually covers, which is primarily based on a single clinical trial. To accurately reflect the study design and provide clarity for readers, we have revised the title as follows:

“Economic evaluation of dialysis treatment in end-stage renal disease patients with fluid and sodium overload: evidence from a randomized controlled trial in Thailand”

• Although mortality and utility parameters from the Singapore study are included in the sensitivity analysis, presenting these findings probabilistically—such as through cost-effectiveness acceptability curves—would be more informative than relying solely on deterministic ICERs.

Response: Thank you very much for your valuable suggestion. In response, we have presented the probabilistic sensitivity analysis (PSA) findings using cost-effectiveness acceptability curves (CEACs) for both the base-case (Figure 4) and the scenario using mortality and utility parameters from Yang et al.’s study in Singapore (Figure 5), thereby providing a more informative representation of decision uncertainty. Additional explanations have been included in the Results section (page 15-16, lines 316-320)

“At Thailand’s WTP threshold of 160,000 THB (4,603 USD) per QALY gained, glucose-based CAPD had the highest probability of being cost-effective (87%), followed by APD (13%). When applying mortality and utility parameters from Yang et al.’s study, CAPD remained dominant with a 97% probability of cost-effectiveness, reinforcing the robustness of our conclusions across parameter variations.”

• While I appreciate the additional discussion on the current dialysis policy in Thailand, please clarify the following sentence in a more explicit manner: "Our study's findings must also be interpreted in light of the recent policy shift toward an APD-first approach implemented by the NHSO in 2024. While this may affect the direct applicability of our results,...." Specifically, it would be helpful to elaborate on how this policy change might influence the relevance of your findings, as well as how your study’s results can inform or relate to the current policy, taking into account differences in patient populations, treatment pathways, or other relevant factors.

Response: Thank you for your valuable comment. We have revised the manuscript to clarify how the APD-first policy introduced by the NHSO in 2024 may influence the interpretation of our study findings and how our results can inform current policy discussions (Discussion, page 22-23, lines 464-477).

“Our study's findings must also be interpreted in light of the recent policy shift toward an APD-first approach implemented by the NHSO in 2024. Although CAPD was identified as the most cost-effective option under Thailand’s WTP threshold, and APD was found to be dominated in the subgroup of patients with fluid and sodium overload, the APD-first policy reflects broader considerations beyond cost-effectiveness, including patient preference, service logistics, and workforce distribution. Importantly, the Thai HTA Guideline [14] emphasizes that policy decisions should not be based solely on cost-effectiveness but should also incorporate ethical implications, equity concerns, and social values. In this context, APD may not be cost-effective in certain subgroups but could still be justified for patients unable to perform CAPD independently or for those requiring greater flexibility to accommodate work or caregiving responsibilities. Thus, our findings remain relevant for informing refinement of the APD-first policy by identifying clinical contexts and patient subgroups where APD use is both clinically appropriate and economically reasonable, thereby supporting Thailand’s goals of universal health coverage, patient-centered care, and equitable resource allocation.”

Reviewer #2:

Thank you for addressing my suggestion. Table S4 provides a clear and useful summary of the baseline characteristics and enhances the interpretability of the study. However, I recommend that the authors explicitly cite this supplementary table in the main text, ideally in the Methods section when describing the study population.

With this minor revision, I have no further concerns regarding the manuscript.

Response: Thank you for your valuable comment. We have cited Table S4 under Methods section (page 5, lines 101-102).

---

## [Editor Report · Decision Letter 3]

16 Oct 2025

Economic evaluation of dialysis treatment in end-stage renal disease patients with fluid and sodium overload: evidence from a randomized controlled trial in Thailand

PONE-D-25-05632R3

Dear Dr. Chaikledkaew,

We’re pleased to inform you that your manuscript has been judged scientifically suitable for publication and will be formally accepted for publication once it meets all outstanding technical requirements.

Kind regards,

Teerawat Thanachayanont

Academic Editor

PLOS ONE
---

## [Editor Report · Acceptance letter]

PONE-D-25-05632R3

PLOS ONE

Dear Dr. Chaikledkaew,

I'm pleased to inform you that your manuscript has been deemed suitable for publication in PLOS ONE. Congratulations! Your manuscript is now being handed over to our production team.

Kind regards,

on behalf of

Dr. Teerawat Thanachayanont

Academic Editor

PLOS ONE